# Penning micro-trap for quantum computing

Shreyans Jain[1,2,5 ✉], Tobias Sägesser[1,2,5], Pavel Hrmo[1,2], Celeste Torkzaban[1], Martin Stadler[1,2], Robin Oswald[1,2], Chris Axline[1], Amado Bautista-Salvador[3,4], Christian Ospelkaus[3,4], Daniel Kienzler[1,2] & Jonathan Home[1,2]

Trapped ions in radio-frequency traps are among the leading approaches for realizing quantum computers, because of high-fidelity quantum gates and long coherence times[1–3]. However, the use of radio-frequencies presents several challenges to scaling, including requiring compatibility of chips with high voltages[4], managing power dissipation[5] and restricting transport and placement of ions[6]. Here we realize a micro-fabricated Penning ion trap that removes these restrictions by replacing the radio-frequency field with a 3 T magnetic field. We demonstrate full quantum control of an ion in this setting, as well as the ability to transport the ion arbitrarily in the trapping plane above the chip. This unique feature of the Penning micro-trap approach opens up a modification of the quantum charge-coupled device architecture with improved connectivity and flexibility, facilitating the realization of large-scale trapped-ion quantum computing, quantum simulation and quantum sensing.

Trapped atomic ions are among the most advanced technologies for realizing quantum computation and quantum simulation, based on a combination of high-fidelity quantum gates[1–3] and long coherence times[7]. These have been used to realize small-scale quantum algorithms and quantum error correction protocols. However, scaling the system size to support orders-of-magnitude more qubits[8,9] seems highly challenging[10–13]. One of the primary paths to scaling is the quantum charge-coupled device (QCCD) architecture, which involves arrays of trapping zones between which ions are shuttled during algorithms[13–16]. However, challenges arise because of the intrinsic nature of the radio-frequency (rf) fields, which require specialized junctions for two-dimensional (2D) connectivity of different regions of the trap. Although successful demonstrations of junctions have been performed, these require dedicated large-footprint regions of the chip that limit trap density[17–21]. This adds to several other undesirable features of the rf drive that make micro-trap arrays difficult to operate[6], including substantial power dissipation due to the currents flowing in the electrodes, and the need to co-align the rf and static potentials of the trap to minimize micromotion, which affects gate operations[22,23]. Power dissipation is likely to be a very severe constraint in trap arrays of more than 100 sites[5,23].

An alternative to rf electric fields for radial confinement is to use a Penning trap in which only static electric and magnetic fields are used, which is an extremely attractive feature for scaling because of the lack of power dissipation and geometrical restrictions on the placement of ions[23,24]. Penning traps are a well-established tool for precision spectroscopy with small numbers of ions[25–28], whereas quantum simulations and quantum control have been demonstrated in crystals of more than 100 ions[29–31]. However, the single trap site used in these approaches does not provide the flexibility and scalability necessary for large-scale quantum computing.

Invoking the idea of the QCCD architecture, the Penning QCCD can be envisioned as a scalable approach, in which a micro-fabricated electrode structure enables the trapping of ions at many individual trapping sites, which can be actively reconfigured during the algorithm by changing the electric potential. Beyond the static arrays considered in previous work[23,32], here we conceptualize that ions in separated sites are brought close to each other to use the Coulomb interaction for two-qubit gate protocols implemented through applied laser or microwave fields[33,34], before being transported to additional locations for further operations. The main advantage of this approach is that the transport of ions can be performed in three dimensions almost arbitrarily without the need for specialized junctions, enabling flexible and deterministic reconfiguration of the array with low spatial overhead.

In this study, we demonstrate the fundamental building block of such an array by trapping a single ion in a cryogenic micro-fabricated surface-electrode Penning trap. We demonstrate quantum control of its spin and motional degrees of freedom and measure a heating rate lower than in any comparably sized rf trap. We use this system to demonstrate flexible 2D transport of ions above the electrode plane with negligible heating of the motional state. This provides a key ingredient for scaling based on the Penning ion-trap QCCD architecture.

The experimental setup involves a single beryllium ($^9$Be$^+$) ion confined using a static quadrupolar electric potential generated by applying voltages to the electrodes of a surface-electrode trap with geometry shown in Fig. 1a–c. We use a radially symmetric potential $V(x,y,z) = m\omega_z^2(z^2 - (x^2+y^2)/2)/(2e)$, centred at a position 152 μm above the chip surface. Here, $m$ is the mass of the ion, $\omega_z$ is the axial frequency and $e$ is the elementary charge. The trap is embedded in a homogeneous magnetic field aligned along the $z$-axis with a magnitude of $B \simeq 3$ T, supplied by a superconducting magnet. The trap assembly is placed in a cryogenic, ultrahigh vacuum chamber that fits inside the magnet bore, with the aim of reducing background-gas collisions and motional heating. Using a laser at 235 nm, we load the trap by resonance-enhanced multiphoton ionization of neutral atoms produced from either a resistively heated oven or an ablation source[35]. We regularly trap single ions for more than a day, with the primary loss mechanism

[1]Department of Physics, ETH Zürich, Zurich, Switzerland. [2]Quantum Center, ETH Zürich, Zurich, Switzerland. [3]Institut für Quantenoptik, Leibniz Universität Hannover, Hannover, Germany. [4]Physikalisch-Technische Bundesanstalt, Braunschweig, Germany. [5]These authors contributed equally: Shreyans Jain, Tobias Sägesser. ✉e-mail: sjain@phys.ethz.ch

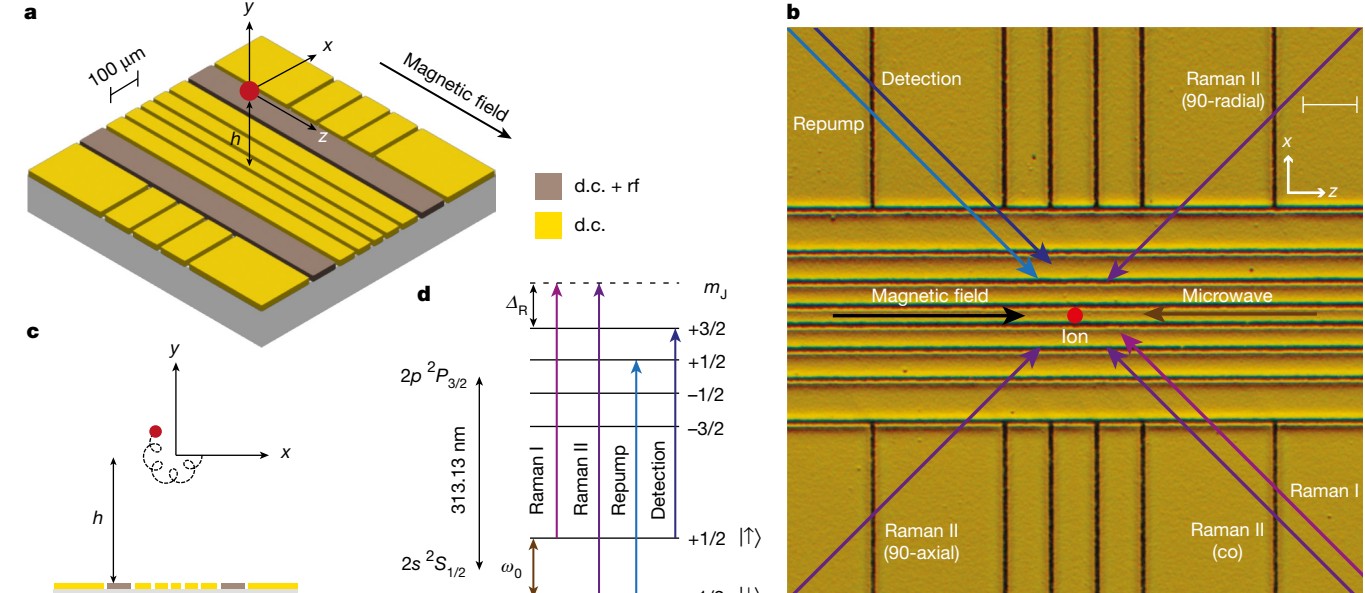

**Fig. 1 | Surface-electrode Penning trap. a**, Schematic showing the middle section of the micro-fabricated surface-electrode trap. The trap chip is embedded in a uniform magnetic field along the $z$ axis, and the application of d.c. voltages on the electrodes leads to 3D confinement of the ion at a height $h \simeq 152\,\mu m$ above the surface. Electrodes labelled 'd.c. + rf' are used for coupling the radial modes during Doppler cooling. **b**, Micrographic image of the trap chip, with an overlay of the direction of the laser beams (all near 313 nm) and microwave radiation (near $\omega_0 \simeq 2\pi \times 83.2$ GHz) required for manipulating the spin and motion of the ion. All laser beams run parallel to the surface of the trap

and are switched on or off using acousto-optic modulators, whereas microwave radiation is delivered to the ion by a horn antenna close to the chip. Scale bar, 100 μm. **c**, Epicyclic motion of the ion in the radial plane ($x$–$y$) resulting from the sum of the two circular eigenmodes, the cyclotron and the magnetron modes. **d**, Electronic structure of the $^9Be^+$ ion, with the relevant transitions used for coherent and incoherent operations on the ion. Only the levels with nuclear spin $m_I = +3/2$ are shown. The virtual level (dashed line) used for Raman excitation is detuned $\Delta_R \simeq +2\pi \times 150$ GHz from the $2p\,^2P_{3/2}\,|m_I = +3/2, m_J = +3/2\rangle$ state.

being related to user interference. Further details about the apparatus can be found in the Methods.

The three-dimensional (3D) motion of an ion in a Penning trap can be described as a sum of three harmonic eigenmodes. The axial motion along $z$ is a simple harmonic oscillator with frequency $\omega_z$. The radial motion is composed of modified-cyclotron ($\omega_+$) and magnetron ($\omega_-$) components, with frequencies $\omega_\pm = \omega_c/2 \pm \Omega$, where $\Omega = \sqrt{\omega_c^2 - 2\omega_z^2}/2$ (ref. 36) and $\omega_c = eB/m \simeq 2\pi \times 5.12$ MHz is the bare cyclotron frequency. Voltage control over the d.c. electrodes of the trap enables the axial frequency to be set to any value up to the stability limit, $\omega_z \leq \omega_c/\sqrt{2} \simeq 2\pi \times 3.62$ MHz. This corresponds to a range $0 \leq \omega_- \leq 2\pi \times 2.56$ MHz and $2\pi \times 2.56$ MHz $\leq \omega_+ \leq 2\pi \times 5.12$ MHz for the magnetron and modified-cyclotron modes, respectively. Doppler cooling of the magnetron mode, which has a negative total energy, is achieved using a weak axialization rf quadrupolar electric field (less than 60 mV peak-to-peak voltage on the electrodes) at the bare cyclotron frequency, which resonantly couples the magnetron and modified-cyclotron motions[37,38]. For the wiring configuration used in this work, the null of the rf field is produced at a height $h \simeq 152\,\mu m$ above the electrode plane. Aligning the null of the d.c. (trapping) field to the rf null is beneficial because it reduces the driven radial motion at the axialization frequency; nevertheless, we find that Doppler cooling works with a relative displacement of tens of micrometres between the d.c. and rf nulls, albeit with lower efficiency. The rf field is required only during Doppler cooling, and not, for instance, during coherent operations on the spin or motion of the ion. All measurements in this work are taken at an axial frequency $\omega_z \simeq 2\pi \times 2.5$ MHz, unless stated otherwise. The corresponding radial frequencies are $\omega_+ \simeq 2\pi \times 4.41$ MHz and $\omega_- \simeq 2\pi \times 0.71$ MHz.

Figure 1d shows the electronic structure of the beryllium ion along with the transitions relevant to this work. We use an electron spin qubit (consisting of the $|\uparrow\rangle \equiv |m_I = +3/2, m_J = +1/2\rangle$ and $|\downarrow\rangle \equiv |m_I = +3/2, m_J = -1/2\rangle$

eigenstates within the $2s\,^2S_{1/2}$ ground-state manifold), which in the high field is almost decoupled from the nuclear spin. The qubit frequency is $\omega_0 \simeq 2\pi \times 83.2$ GHz. Doppler cooling is performed using the detection laser red-detuned from the (bright) $|\uparrow\rangle \leftrightarrow 2p\,^2P_{3/2}\,|m_I = +3/2, m_J = +3/2\rangle$ cycling transition, whereas an additional repump laser optically pumps population from the (dark) $|\downarrow\rangle$ level to the higher energy $|\uparrow\rangle$ level through the fast-decaying $2p\,^2P_{3/2}\,|m_I = +3/2, m_J = +1/2\rangle$ excited state. State-dependent fluorescence with the detection laser allows for discrimination between the two qubit states based on photon counts collected on a photomultiplier tube using an imaging system that uses a 0.55 NA Schwarzschild objective. The fluorescence can also be sent to an electron-multiplying CCD (EMCCD) camera.

Coherent operations on the spin and motional degrees of freedom of the ion are performed either using stimulated Raman transitions with a pair of lasers tuned to 150 GHz above the $2p\,^2P_{3/2}\,|m_I = +3/2, m_J = +3/2\rangle$ state or using a microwave field. The former requires the use of two 313 nm lasers phase-locked at the qubit frequency, which we achieve using the method outlined in ref. 39. By choosing different orientations of Raman laser paths, we can address the radial or axial motions, or implement single-qubit rotations using a co-propagating Raman beam pair.

The qubit transition has a sensitivity of 28 GHz $T^{-1}$ to the magnetic field, meaning the phase-coherence of our qubit is susceptible to temporal fluctuations or spatial gradients of the field across the extent of the motion of the ion. Using Ramsey spectroscopy, we measure a coherence time of 1.9(2) ms with the Raman beams. Similar values are measured with the microwave field, indicating that laser phase noise from beam path fluctuations or imperfect phase-locking does not significantly contribute to dephasing. The nature of the noise seems to be slow on the timescale (about 1 ms to 10 ms) of a single experimental shot consisting of cooling, probing and detection, and the fringe contrast decay follows a Gaussian curve. We note that the coherence

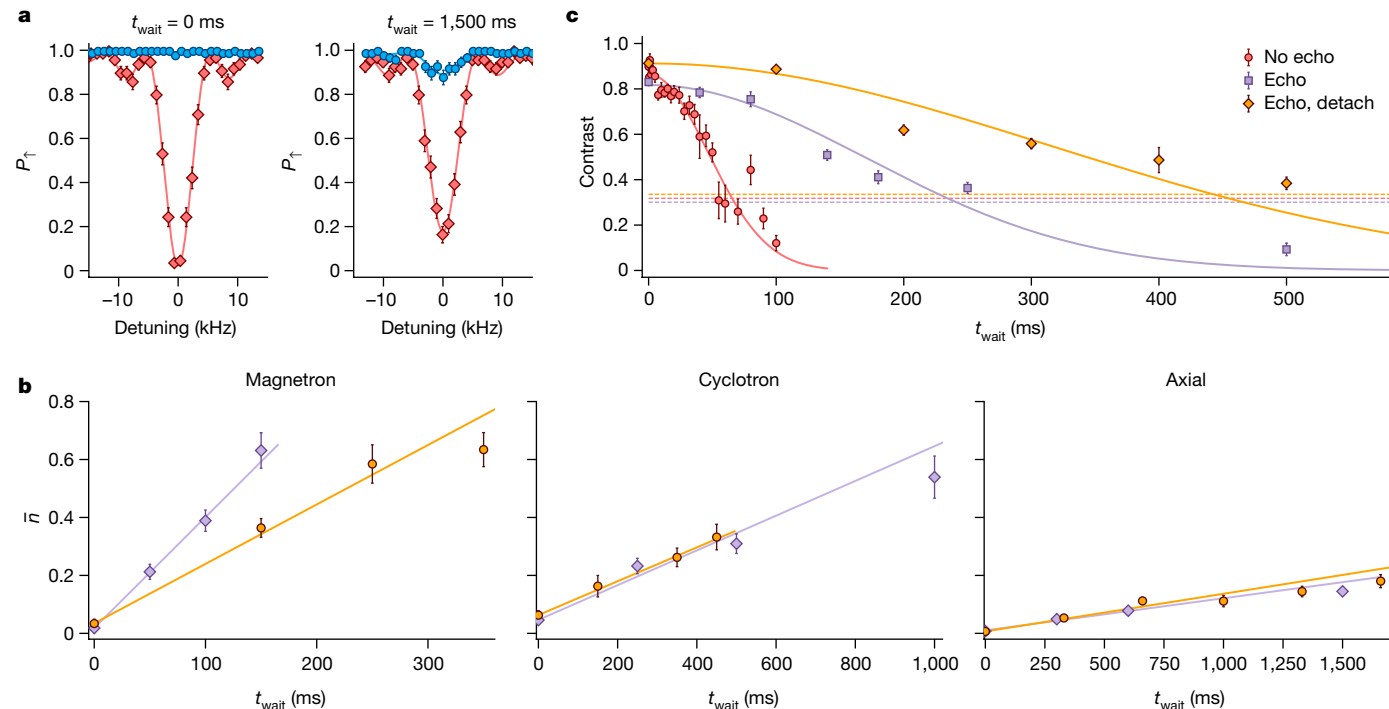

**Fig. 2 | Motional coherence. a**, Bright-state population $P_\uparrow$ measured after applying the first red or blue axial sideband probe-pulse to the sideband-cooled ion. As the bright state $|\uparrow\rangle$ has a higher energy than the dark state $|\downarrow\rangle$, the blue sideband cannot be driven when the ion is in the ground state of the axial mode. **b**, Average phonon number $\bar{n}$ calculated using the sideband-ratio method[44] for all three modes as a function of increasing $t_{wait}$. The purple and orange points indicate data taken with the trap connected and detached, respectively. The heating rates are extracted from the slopes of the linear fits. **c**, Motional

dephasing of the axial mode observed by Ramsey spectroscopy. The purple points indicate data taken with an echo pulse in the sequence. The orange points indicate data taken with an echo pulse, in which, additionally, the trap was detached between Doppler cooling and the detection pulse. Whereas the dataset with the voltage sources detached is taken at $\omega_z \simeq 2\pi \times 2.5$ MHz, the two data series with the trap attached are taken at an axial mode frequency $\omega_z \simeq 2\pi \times 3.1$ MHz. The dashed lines show the $1/e$ line normalized to the Gaussian fits. All error bars indicate the standard error.

is reduced if vibrations induced by the cryocoolers used to cool the magnet and the vacuum apparatus are not well decoupled from the experimental setup. Further characterization of the magnetic field noise is performed by applying different orders of the Uhrig dynamical decoupling sequence[40,41], with the resulting extracted coherence time from the measurements being 3.2(1) ms, 5.8(3) ms and 8.0(7) ms for orders 1, 3 and 5, respectively. Data on spin-dephasing are presented in Extended Data Fig. 1.

A combination of the Doppler cooling and repump lasers prepares the ion in the $|\uparrow\rangle$ electronic state and a thermal distribution of motional Fock states. After Doppler cooling using the axialization technique, we measure mean occupations of $\{\bar{n}_+, \bar{n}_-, \bar{n}_z\} = \{6.7(4), 9.9(6), 4.4(1)\}$ using sideband spectroscopy on the first four red and blue sidebands[38]. Pulses of continuous sideband cooling[31,38] are subsequently performed by alternatively driving the first and third blue sidebands of a positive energy motional mode and red sidebands of a negative energy motional mode while simultaneously repumping the spin state to the bright state. The 3D ground state can be prepared by applying this sequence for each of the three modes in succession. The use of the third sideband is motivated by the high Lamb–Dicke parameters of approximately 0.4 in our system[42,43]. After a total time of 60 ms of cooling, we probe the temperature using sideband spectroscopy on the first blue and red sidebands[44]. Assuming thermal distributions, we measure $\{\bar{n}_+, \bar{n}_-, \bar{n}_z\} = \{0.05(1), 0.03(2), 0.007(3)\}$. We have achieved similar performance of the ground-state cooling at all trap frequencies probed to date. The long duration of the sideband cooling sequence stems from the large (estimated as 80 μm) Gaussian beam radius of the Raman beams each with power in the range of 2 mW to 6 mW, leading to a Rabi frequency $\Omega_0 \simeq 2\pi \times 8$ kHz, which corresponds to π times of

approximately 62 μs, 145 μs and 2,000 μs for the ground-state carrier, first and third sidebands, respectively, at $\omega_z = 2\pi \times 2.5$ MHz.

Trapped-ion quantum computing uses the collective motion of the ions for multi-qubit gates and thus requires the motional degree of freedom to retain coherence over the timescale of the operation[33,45]. A contribution to decoherence comes from motional heating due to fluctuations in the electric field at frequencies close to the oscillation frequencies of the ion. We measure this by inserting a variable-length delay $t_{wait}$ between the end of sideband cooling and the temperature probe. As shown in Fig. 2, we observe motional heating rates $\{\dot{\bar{n}}_+, \dot{\bar{n}}_-, \dot{\bar{n}}_z\} = \{0.49(5)$ s$^{-1}$, $3.8(1)$ s$^{-1}$, $0.088(9)$ s$^{-1}\}$. The corresponding electric-field spectral noise density for the axial mode, $S_E = 4\hbar m\omega_z\dot{\bar{n}}_z/e^2 = 3.4(3) \times 10^{-16}$ V$^2$m$^{-2}$Hz$^{-1}$, is lower than any comparable measurement in a trap of similar size[46,47]. As detailed in the Methods, we can trap ions in our setup with the trap electrodes detached from any external supply voltage except during Doppler cooling, which requires the axialization signal to pass to the trap. Using this method, we measure heating rates $\dot{\bar{n}}_z = 0.10(1)$ s$^{-1}$ and $\dot{\bar{n}}_+ = 0.58(2)$ s$^{-1}$ for the axial and cyclotron modes, respectively, whereas the rate for the lower-frequency magnetron mode drops to $\dot{\bar{n}}_- = 1.8(3)$ s$^{-1}$. This reduction suggests that external electrical noise contributes to the higher magnetron heating rate in the earlier measurements.

Motional-state dephasing was measured using Ramsey spectroscopy, involving setting up a superposition $|\uparrow\rangle (|0\rangle_z + |1\rangle_z)/\sqrt{2}$ of the first two Fock states of the axial mode (here $\omega_z \simeq 2\pi \times 3.1$ MHz) using a combination of carrier and sideband pulses[48]. Following a variable wait time, we reverse the preparation sequence with a shifted phase. The resulting decay of the Ramsey contrast shown in Fig. 2c is much faster than what would be expected from the heating rate. The decay is roughly

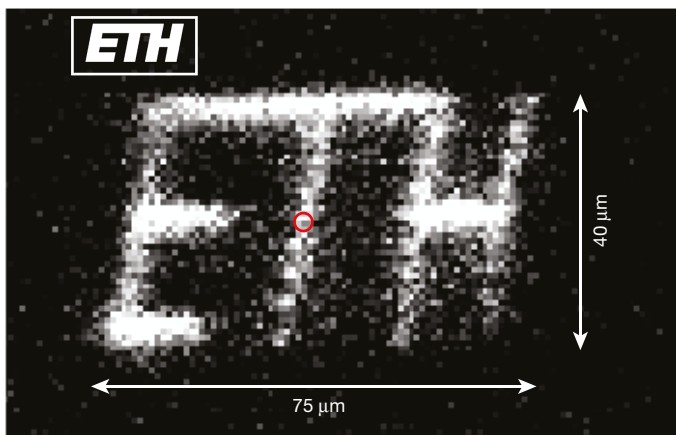

**Fig. 3 | Demonstration of 2D transport.** A single ion is transported adiabatically in the x–z plane (normal to the imaging optical axis). The ion is illuminated for 500 µs at a total of 58 positions, here defined by the ETH Zürich logo (see inset for reference image). The red circle indicates the initial position in which the ion is Doppler-cooled. The ion is moved across a region spanning approximately 40 µm and 75 µm along the x (radial) and z (axial) directions, respectively. The sequence is repeated 172 times to accumulate the image.

Gaussian in form with a 1/e coherence time of 66(5) ms. Inserting an echo pulse in the Ramsey sequence extends the coherence time to 240(20) ms, which indicates low-frequency noise components dominating the bare Ramsey coherence. Further improvement of the echo coherence time to 440(50) ms is observed when the trap electrodes are detached from external voltage sources between the conclusion of Doppler cooling and the start of the detection pulse, in which again the axialization signal is beneficial. The data with the voltage sources detached are taken at $\omega_z \simeq 2\pi \times 2.5$ MHz.

An important component of the QCCD architecture[14] is ion transport. We demonstrate that the Penning trap approach enables us to perform this flexibly in two dimensions by adiabatically transporting a single ion, and observing it at the new location. The ion is first Doppler-cooled at the original location, and then transported in 4 ms to a second desired location along a direct trajectory. We then perform a 500-µs detection pulse without applying axialization and collect the ion fluorescence on an EMCCD camera. The exposure of the camera is limited to the time window defined by the detection pulse. The lack of axialization is important when the ion is sufficiently far from the rf null to minimize radial excitation due to micromotion and subsequently produce enough fluorescence during the detection window. The ion is then returned to the initial location. Figure 3 shows a result in which we have drawn the first letters of the ETH Zürich logo. The image quality and maximum canvas size are only limited by the point-spread function and field of view of our imaging system, as well as the spatial extent of the detection laser beam, and not by any property of the transport. Reliable transport to a set location and back has been performed up to 250 µm. By probing ion temperatures after transport using sideband thermometry (Extended Data Fig. 2), we have observed no evidence of motional excitation from transport compared with the natural heating expected over the duration of the transport. This contrasts with earlier non-adiabatic radial transport of ensembles of ions in Penning traps, in which a good fraction of the ions were lost in each transport[49].

This work marks a starting point for quantum computing and simulation in micro-scale Penning trap 2D arrays. The next main step is to operate with multiple sites of such an array, which will require optimization of the loading while keeping the ions trapped in shallow potentials. This can be accomplished in the current trap with the appropriate wiring, but notable advantages could be gained by using a trap with a loading

region and shuttling ions into the micro-trap region. Multi-qubit gates could then be implemented following the standard methods demonstrated in rf traps[23,34]. Increased spin-coherence times could be achieved through improvements to the mechanical stability of the magnet, or in the longer term through the use of decoherence-free subspaces, which were considered in the original QCCD proposals[14,50,51]. For scaling to large numbers of sites, it is likely that scalable approaches to light delivery will be required, which might necessitate switching to an ion species that is more amenable to integrated optics[52–55]. The use of advanced standard fabrication methods such as CMOS[56,57] is facilitated, compared with rf traps, by the lack of high-voltage rf signals. Compatibility with these technologies demands an evaluation of how close to the surface ions could be operated for quantum computing and will require in-depth studies of heating—here an obvious next step is to sample electric field noise as a function of ion-electrode distance[47]. Unlike in rf traps, 3D scans of electric field noise are possible in any Penning trap because of the flexibility of confinement to the uniform magnetic field. This flexibility of ion placement has advantages in many areas of ion-trap physics, for instance, in placing ions in anti-nodes of optical cavities[58], or sampling field noise from surfaces of interest[59,60]. We, therefore, expect that our work will open previously unknown avenues in sensing, computation, simulation and networking, enabling ion-trap physics to break out beyond its current constraints.

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

# Methods

## Spin-coherence data

The data for spin-dephasing measurements can be found in Extended Data Fig. 1. The $N$th order Uhrig sequence is performed by adding $N$ π-pulses centred at times

$$\tau_n = t_{wait}\sin^2\left(\frac{n\pi}{2(N+1)}\right) \qquad (1)$$

between the two π/2 pulses.

## Trap design and fabrication

The design of our surface-electrode Penning trap is based on the traditional four-wire or five-wire traps widely used in the rf trapped-ion community[61]. Two configurations of arranging ions were considered during the design phase: (1) a single trapping site that can be placed at a variable height above the surface and (2) two trap sites separated along the in-plane radial axis. In total, the trap consists of 25 electrodes laid out on a planar surface and surrounded by a conducting ground plane. All electrodes are supplied with static voltages, whereas an individual rf signal can be applied to each of the middle seven electrodes running parallel to the magnetic field, such that they also potentially serve as axialization electrodes (Fig. 1b). The static voltages applied to the electrodes enable us to control the elements of the Hessian matrix of the trapping potential. Although typical rf traps might use two or three rf electrodes, the extra rf electrodes on our trap allow us to provide two nodal lines with variable separation in the rf axialization potential, or to produce a node at a variable height above the chip surface. Experiments included in this study have been restricted to ions in a single trap site 152 μm above the trap surface, with only the outermost rf electrodes used for axialization. This height is determined using an analytical calculation and confirmed using an independent simulation based on the boundary element method.

The trap was fabricated at PTB, Braunschweig, using a single-layer processing method[62] depositing gold electrodes by electroplating on a sapphire substrate. The process yields an electrode thickness in the range of 10 μm to 12 μm and approximately 3-μm-wide gaps between the electrodes, which provides excellent electrical shielding of the substrate.

## Cryogenic vacuum apparatus

The trap is placed in a cryogenic vacuum apparatus that is inserted into the horizontal magnet bore (Extended Data Fig. 3). The trap enclosure (Extended Data Fig. 4) is surrounded by a vacuum chamber and heat shields and is held at a temperature of 6.5 K. Laser beams for photoionization, Doppler cooling, repumping, state detection and stimulated Raman transitions are delivered along the magnet bore through a vacuum viewport and are directed across the trap by mirrors mounted within the cryogenic trap enclosure.

The detection laser beam is delivered at an angle of 45° with respect to the magnetic field, such that it overlaps all the motional-mode eigenvectors. Three configurations of the Raman beams are possible. The Raman I + Raman II (co) beams are co-propagating and produce a negligible wavevector difference, which causes negligible coupling to the motional modes. The other two beam pairs, Raman I + Raman II (90-axial) and Raman I + Raman II (90-radial) have wavevector differences along the axial and the radial motional modes, respectively. Fluorescence from the ion is collected by a Schwarzschild objective and directed down the bore of the magnet by a further mirror. Microwave radiation is delivered through a hollow WR-10 waveguide and directed across the trap using a horn antenna. An effusive oven is placed at the room temperature stage of the apparatus and generates a flux of neutral $^9$Be atoms, which is directed at the centre of the trap chip.

Ultrahigh vacuum at the trap is achieved by cryopumping. Furthermore, an activated charcoal getter is added within the trap enclosure to improve the helium partial pressure. Vacuum levels in the room temperature stage of the apparatus are measured to be less than $5 \times 10^{-10}$ mbar. Although we cannot directly measure the vacuum pressure within the trap enclosure, we routinely keep the ions for several days with primary loss mechanisms related to equipment malfunction or user error, indicating excellent vacuum.

## Voltage-source detachment

As confinement of an ion in a Penning trap requires only static electric and magnetic fields, it is possible to temporarily detach the trap electrodes from the voltage sources external to the Faraday cage formed by the vacuum chamber. In every line connecting a trap electrode to a digital-to-analog converter and the lab environment, we place switches that can be actuated using a digital signal (Extended Data Fig. 5). In our experimental sequence, we detach the trap after Doppler cooling, which requires the axialization drive to pass to the trap.

The combined capacitance of the trap electrode and the in-vacuum RC filter will retain the voltage that was applied by the digital-to-analog converter before isolating the trap. Although discharging is observed over timescales of minutes, we find no measurable change of ion position or motional-mode frequencies during up to 2 s of trap detachment.

To reach maximal isolation from noise present in the lab environment, we concatenate three MOSFET-based switches (G3VM-41QR10, Omron Electronics). The capacitor $C$ acts as a capacitive divider together with the switch capacitance $C_{off}$ to increase isolation performance over the full range of possible motional-mode frequencies (d.c. to 5.118 MHz). Simulating this circuit and including parasitic capacitances yields an estimated isolation of 83 dB at d.c. and 77 dB at 5 MHz.

The option to entirely isolate the trap from the lab environment during parts of the experimental sequence is exclusive to Penning traps. It may permit reduced filtering requirements, in turn enabling faster ion transport, while still allowing low noise levels when the ions are kept at static positions[63]. It may simplify the task of eliminating external noise sources that are hard to find or hard to remove.

## Data availability

The data supporting the findings of this work are available in the paper, the Extended Data and the ETH Zürich Research Collection repository (open access https://doi.org/10.3929/ethz-b-000627348).

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

**Acknowledgements** This project has received funding from ETH Zürich, the ERC under the Horizon 2020 research and innovation programme of the European Union (EU) (grant agreement no. 818195), the EU Quantum Flagship H2020-FETFLAG-2018-03 (grant agreement no. 820495 AQTION), and the EU H2020 FET Open project PIEDMONS (grant no. 801285). S.J. thanks E. Brucke for assistance in the cleanroom and J. Alonso Otamendi for his involvement in the work building up to the experiment assembly. T.S. thanks P. Clements for designing the trap detachment PCB. A.B.-S. and C.O. thank the cleanroom staff, in particular T. Weimann, P. Hinze and O. Kerker, and acknowledge funding from PTB, QUEST, LUH and DFG through CRC 1227 DQ-mat, project A01. We thank A. Ricci Vasquez and M. Simoni for the careful reading and assessment of the paper.

**Author contributions** Data taking and analysis were performed by S.J., T.S. and P.H. The apparatus was primarily built by T.S. and S.J. with contributions from P.H., C.T., C.A., R.O. and M.S. The trap was fabricated by A.B.-S. with input from C.O. The paper was written by S.J., T.S., P.H. and J.H. with input from all authors. The work was supervised by D.K. and J.H.

**Funding** Open access funding provided by Swiss Federal Institute of Technology Zurich.

**Competing interests** A.B.-S. and C.O. are associated with Qudora Technologies, a commercially oriented quantum computing company. The other authors declare no competing interests.

**Additional information**

**Correspondence and requests for materials** should be addressed to Shreyans Jain.

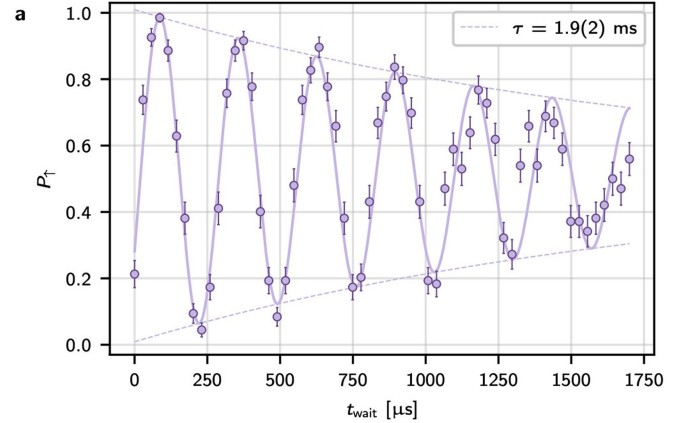

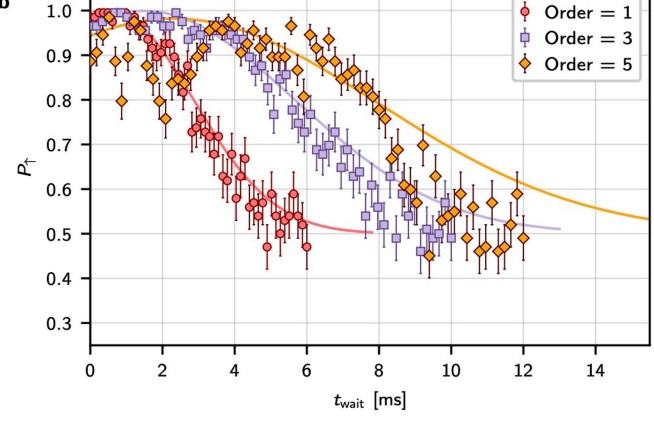

**Extended Data Fig. 1 | Spin coherence.** Population in the $|\uparrow\rangle$ state **a**. as a function of the wait time $t_{wait}$ in the detuned Ramsey experiment, and **b**. for the first three odd orders of the Uhrig dynamical decoupling sequence performed using the Raman beams. In the Uhrig sequence, the phase of the final pulse is chosen to maximise population at 0 wait time. As there is always a $\pi$-pulse in the centre of the wait time, any static detuning offsets are cancelled and the population in the $|\uparrow\rangle$ state is proportional to the contrast. The coherence time is extracted from an empirical fit to an exponential function in **a**. and a Gaussian in **b**. All error bars in this figure indicate the standard error.

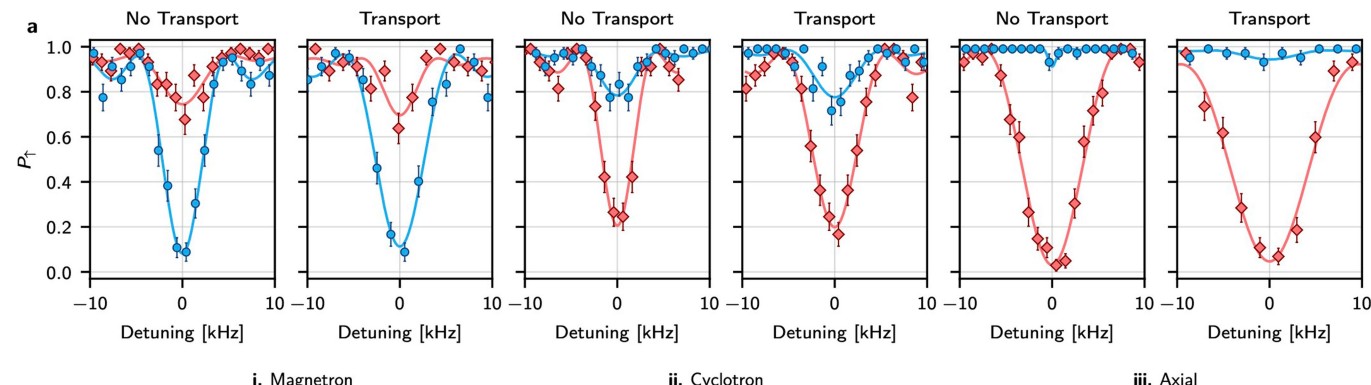

**i.** Magnetron  **ii.** Cyclotron  **iii.** Axial

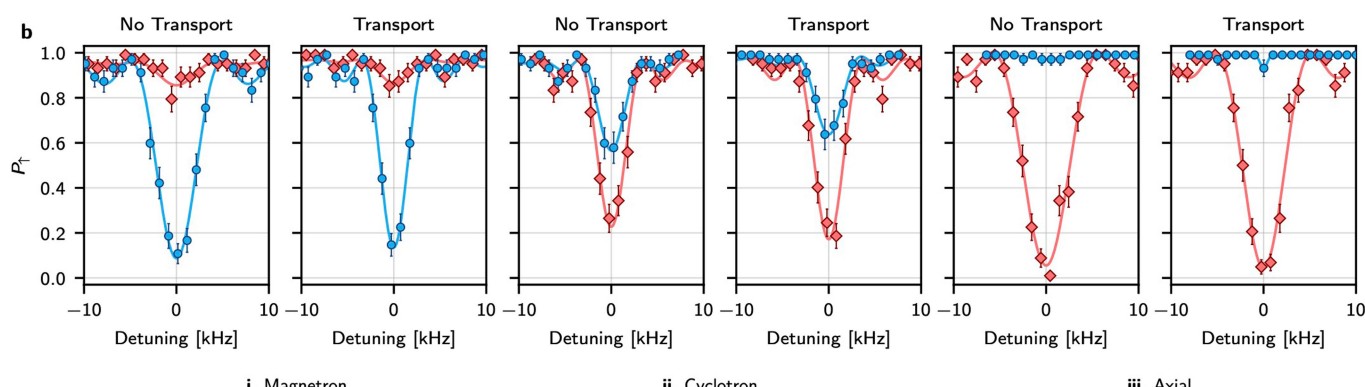

**i.** Magnetron  **ii.** Cyclotron  **iii.** Axial

**Extended Data Fig. 2 | Heating during transport.** Bright-state population $P_\uparrow$ measured after applying the first red or blue axial sideband probe-pulse before and after a number of shuttling events. In panel **a**, the ion is moved along the z-axis (axial direction), while in panel **b**, the ion is moved along the x-axis (in-plane radial direction) - in both cases, the distance moved is approximately 30 μm. For the axial mode, the transport sequence between the initial and final locations is repeated 10 times before the temperature is probed. A single one-way transport sequence has a duration of approximately 300 μs. The temperature is then compared to the case where the ion is held at the initial position for an equivalent total waiting time; this case is labelled as 'No Transport' in the figure. In the case of the magnetron and cyclotron modes, the transport-sequence is repeated 4 times instead. We see no appreciable change in the ion temperature when the transport time is increased. All error bars in this figure indicate the standard error.

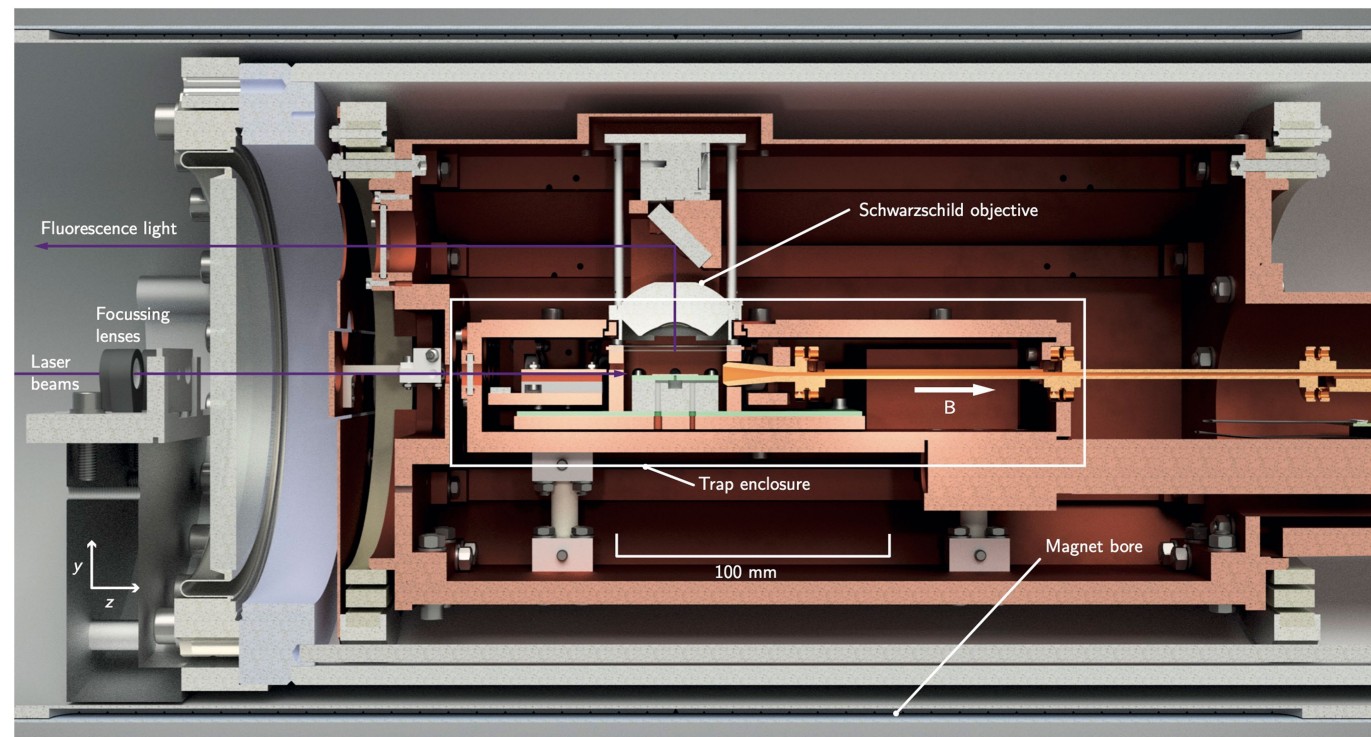

**Extended Data Fig. 3 | Cryogenic vacuum apparatus.** Rendering of the cryogenic vacuum apparatus placed inside the magnet bore. Laser light is delivered along the bore to the trap enclosure, passing focusing lenses and a vacuum viewport. Ion fluorescence is collected using a Schwarzschild objective and directed out of the apparatus and the magnet bore using a mirror.

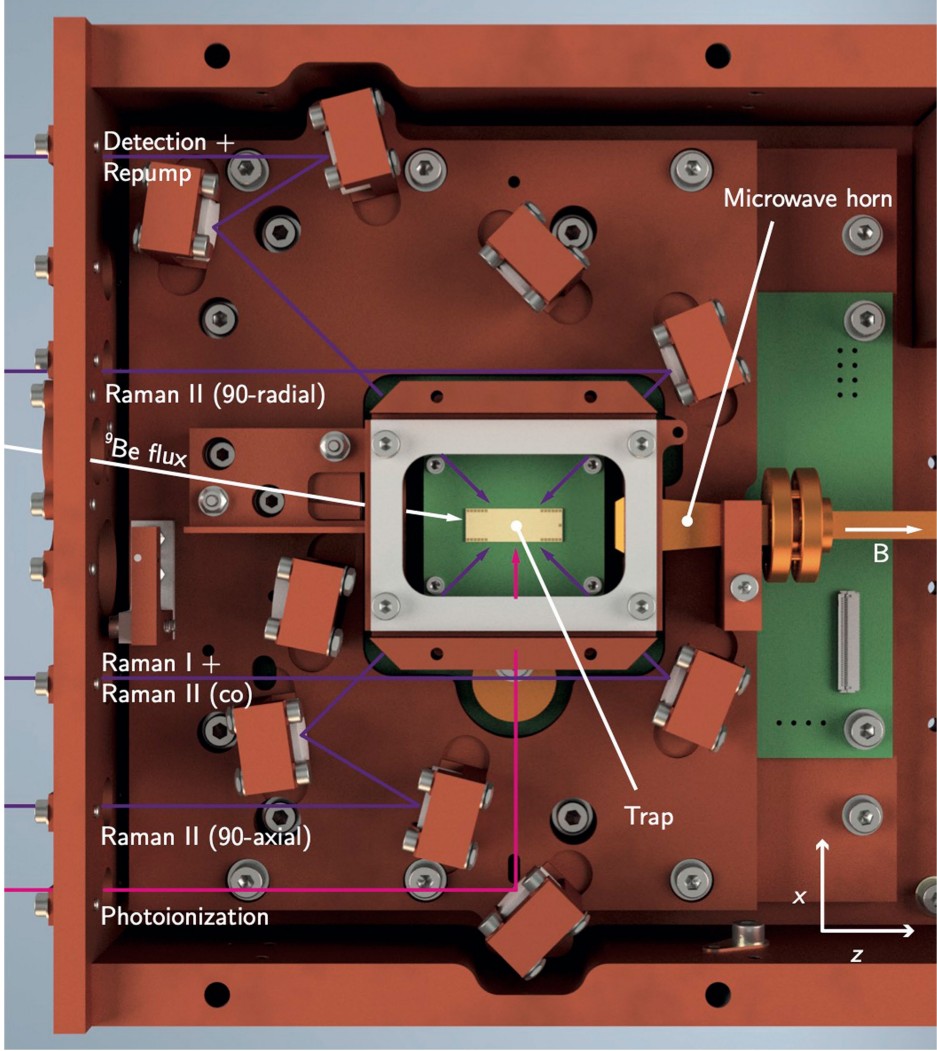

**Extended Data Fig. 4 | Trap enclosure.** Top view of the trap enclosure with the Schwarzschild objective and lid removed. The laser beams are directed across the trap chip by an array of mirrors. A horn antenna delivers microwave radiation across the trap.

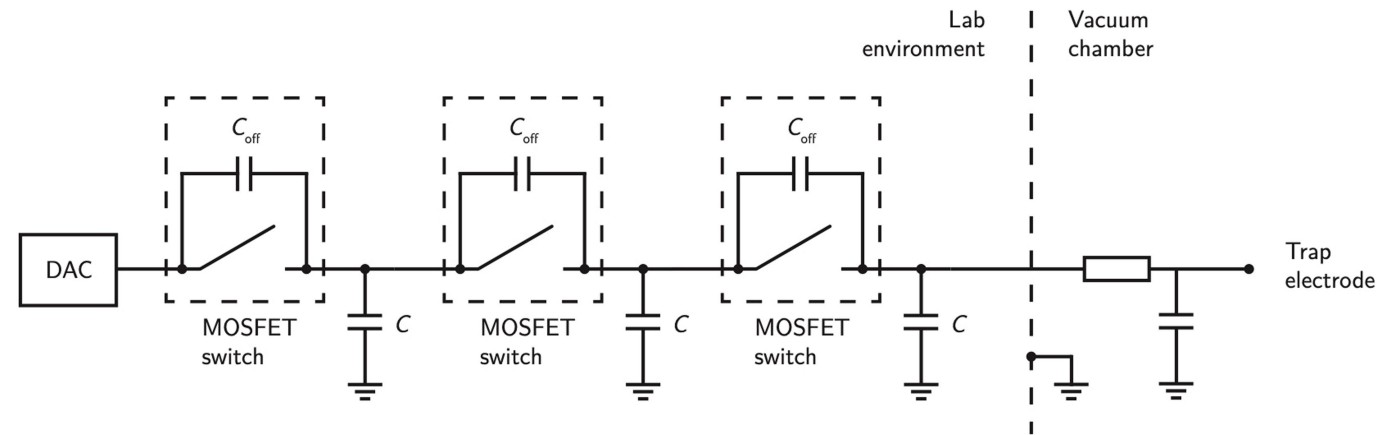

**Extended Data Fig. 5 | Trap detachment.** Schematic of the switch arrangement used to detach a trap electrode from the external DAC. Three MOSFET-based switches with an on-state resistance of $11\,\Omega$ and off-state capacitance $C_{off} = 0.45$ pF are chained and interleaved with capacitors $C = 560$ pF, forming capacitive dividers for improved isolation at radio frequencies. The voltage last applied by the DAC is retained by the combined capacitance of the trap electrode and an RC filter ($R = 1\,k\Omega$, $C = 560$ pF) placed within the vacuum apparatus.