## [Peer Review File · Nature]

Manuscript Title: Penning micro-trap for quantum computing

Reviewer Comments & Author Rebuttals

Reviewer Reports on the Initial Version:

Referees' comments:

Referee #1 (Remarks to the Author):

I think this is an excellent article that reports a significant advance in the field of quantum information processing with trapped ions. This is not the first time that a single ion in a Penning trap has been trapped, sideband cooled and manipulated using quantum control techniques, but it is the first time that this has been done with a microfabricated Penning trap that has the potential for scaling to large numbers of ions by fabricating an array of such traps. The trap design is elegant and novel, and the techniques that are used are advanced and highly effective. This work represents a very significant advance on earlier work with ions in a Penning trap. The low motional excitation after sideband cooling in 3-d and the very low heating rates demonstrated are impressive and advance significantly on previous work.

The article is clear and informative. It has an appropriate set of references. The data is convincing and the figures are helpful. The methods are sound.

I have one significant criticism. The authors claim that they demonstrate "the ability to transport the ion arbitrarily in the trapping plane" (abstract). They claim that figure 3 demonstrates shuttling of the ion. My understanding of "shuttling" would be moving an ion between different trapping zones and this is a requirement for quantum information processing using trapped ions, as described by the authors on page 1. What the authors have done here is to move an ion around within one trapping zone, with a maximum excursion of about 40um (figure 3). This type of movement is not much more advanced than adjusting the position of the ion in order to minimise micromotion, which is a commonly-used technique (e.g. line 113). The range of movement demonstrated here is not much more than the width of one electrode. I think that this claim is therefore too strong and is not justified by the results presented. In order to be able to claim arbitrary transport of an ion, the authors should be able to show that they have moved the ion between different trapping zones.

In the abstract it is not clear what they mean by "This unique feature..". If this relates to their claimed ability to transport the ion, it is unjustified. What is it that's "unique"? Transport in 2-d has been demonstrated many times with microfabricated RF traps. Where have they demonstrated "improved connectivity and flexibility" in this paper?

In line 23 the authors imply that their Penning trap design either eliminates the need for specialized junctions or makes them easier to construct, and I do not believe that this is the case. In this paper they don't even discuss how to move an ion around on a quantum CCD, let alone through junctions.

If they want to make this claim, I think the justification needs to be clearly explained.

In line 54, the static array references should include Crick et al, Rev Sci Instrum 81, 013111 (2010), where shuttling between different trapping sites was demonstrated in a Penning trap. This discussion about ions in separate sites being brought together, and arbitrary transport in 3-d, is way beyond what is demonstrated in this paper.

As discussed above, the claim in lines 69-72 that they have demonstrated the "key ingredient for scaling" is not justified in my view. This may be possible in the future but there are significant challenges to overcome first.

In figure 1a it would be useful to give an indication of scale.

In line 138 it would be useful to know how many photons are detected before the ion decays to the dark level. How long does this take?

In line 165 it would be helpful to define what is meant by a "single experiment" and to state what that timescale is. Also, what is the repetition rate of experiments?

In the caption of figure 2 I suggest including the axial frequency for the experiment with the trap detached in the caption, to save the reader having to look elsewhere in the paper.

The statement in line 276-9 is not very convincing as the ion is only Doppler cooled here. A more convincing and sensitive demonstration would be moving a ground-state cooled ion from one place to another and then demonstrating no heating of the motion during this process.

A small point about the references: there are many examples of missing capital letters (especially for Penning) and lost formatting of super- and sub-scripts.

I find the exceedingly brief statement in lines 620-621 and the two-line caption inadequate for Fig S1. The authors should explain what the data is showing and how it was achieved. Is part b a plot of fringe visibility or population?

In line 631 it is frustrating that they state there are 25 electrodes but we can only see 21 in figure 1. Why not show them all?

Somewhere it would be good to state how the photo-ionization is achieved. Which lasers are used and what is the atomic process?

Referee #2 (Remarks to the Author):

* Summary of the manuscript

Typical approach for the construction of quantum computer (QC) with trapped ions utilizes micro-fabricated Paul trap which requires rf fields with high voltage. So far, quantum computation and quantum simulation with tens of ions (qubits) based on this approach have been experimentally demonstrated already, and they even lead to many commercialized systems. However, to increase the number of qubits beyond the current scale, several limitations should be addressed.

Some of the remaining challenges are the significant power dissipation originating from the large rf voltages necessary for all Paul traps, and the accompanying micromotion which requires frequent re-alignment of the DC null against the rf null. The authors claim that the requirement of rf fields can be resolved by using a Penning trap which mainly requires static magnetic and electric fields only, and they demonstrated that a single 9Be^+ ion can be trapped in a micro-fabricated Penning trap, and three motional modes of this ion can be cooled down very close to the ideal ground states. One of the challenges in reaching the motional ground states of the Penning trap is the Doppler cooling of the magnetron mode with a negative quantized energy, and such mode has been efficiently cooled by the axialization process that mixes the magnetron mode and the cyclotron mode with small amplitude of rf voltage, and then through an additional sideband cooling. They also characterized the coherence time of axial modes and the internal electronic state which will be used as qubit. The measurement result also showed that the significant part of the motional dephasing comes from the electrical noise originating from the external electronics, especially for the magnetron mode having relatively lower frequency. Finally, they also claim that the QCCD architecture originally devised for the micro-fabricated Paul trap can be extended to the micro-fabricated Penning trap by demonstrating the adiabatic shuttling of a single ion, which in principle has more degrees of freedom compared to the Paul trap where shuttling is considered only along the rf rails.

The main novelty of this work is that the authors demonstrated many of components necessary to build scalable quantum computing systems with trapped ions using Penning trap, and provided convincing evidences that micro-fabricated Penning trap might be a strong alternative to the current Paul trap approach. They even devised a novel method to detach the external voltage sources in the middle of quantum operations, which can allow low-noise operation and measurement and might even find applications in other systems. On the other hand, this work did not demonstrate any components required for multi-qubit operation such as trapping of the controlled numbers of ions, combining and splitting of them in the surface trap, and quantum gate between two qubits etc. I should admit that implementation of these components will require fair amount of additional work and each of them can even lead to a separate publication, and therefore it might be still too much to require results of any of these operations in the current manuscript. However, this also might mean that implementation of these components will be non-trivial for the experiments. In such a case, I think it will be fair for the authors to elaborate more about the remaining challenges to demonstrate the multi-qubit operations in the conclusions.

I find that the overall manuscript is well-written and what has been done experimentally is properly

presented. The authors used standard method to estimate the number of phonons and heating rate in each motional mode, and Ramsey interference has been utilized to measure the coherence time of both spin and motional mode while dynamic decoupling methods have been applied to increase the coherence time. Therefore, all the included data and methodology are valid for the purpose of the experiments. Also, all the relevant references are properly included.

* Major concerns

- None

* Minor concerns

- I wonder whether the expression of "unit cell" in the title is appropriate. "Unit cell" can imply that the entire system can be constructed only by replication of the unit cells, but given that the multi-qubit operations are missing, the title might be misleading.

- Similar to the above point, the claim that "we demonstrate full quantum control of an ion" in the abstract needs some contemplation, because "full quantum control of an ion" might mean that the quantum state of the motional modes is also fully controlled such as the implementation of the state dependent force or generation of Schrodinger cat state as shown in PRL 116, 140402 (2016) by the same group, but this work mostly includes partially incoherent process such as sideband cooling and phonon number estimation. Of course, the sideband Ramsey experiment in Fig. 2 (c) might be considered as coherent control of the motional mode to some degree, but with the coherent control, there should be more characterization of the fidelity of the operation. Also, people will be more interested in the application of these states to multi-qubit gates, and given that the two-qubit operations are missing, the use of "full quantum control of an ion" can be misleading.

- Considering the ion height of the 152um and the axial frequency of 2.5 MHz, I am a little worried about the necessary DC voltages for the axial confinement, but I could not find the voltage information in the manuscript. Especially, in the conclusion, the authors claim that the fabrication of the planar Penning trap will be compatible with the CMOS fabrication process, but if the necessary DC voltages are in the range of tens of volts, it might not be compatible with CMOS foundry. Therefore, I want the authors to include more information about the DC voltages. Also, as the authors mentioned in the conclusion, incorporation of the integrated optics for scalability might require use of heavier ions such as Sr⁺ or Ba⁺, and I am afraid that other challenges might creep up as higher DC voltage might be necessary to maintain the similar axial frequency. Especially, for the two-qubit operation, if they plan to apply schemes based on the axial phonon modes, I believe that they want to keep the axial frequency high as in the current setup, but I might be wrong. I hope the authors give more comments on the use of heavier ions.

- In the conclusion, coherence time of spin can be increased by the improved mechanical stability of magnet, but there is no comment on the fluctuation of the current. Could you please include estimated error caused by the mechanical instability and the fluctuation of the current?

- To generate Fig. 3, the ion was Doppler cooled at the original location, but there is no information

about this location. Please indicate where this original location is in Fig. 3. Also, Fig. 3 shows thicker horizontal lines compared to the slightly tilted vertical lines, and I wonder if this is caused by visual artifact, or there is any physical explanation.

- Electrical detachment of the trap from the external voltage source seems to be a useful trick for other experiments, and many readers might find useful if you can include the model number of the MOSFET switch.

* Conclusions

The experimental results of this study provide significant evidence that the micro-fabricated Penning trap might open a new avenue towards scalable quantum computing architecture with trapped ions, even though it seems that there remain many challenges. And, publication of this work can motivate more researchers to attack these challenges, and expedite the overall progress in the development of quantum computer based on ion trap.

Referee #3 (Remarks to the Author):

A. This manuscript describes the operation of a microfabricated Penning trap and full quantum control of the ion in the trap. The authors correctly note that Penning traps have advantages over Paul traps in terms of reduced voltage requirements, lower power dissipation, ion density, and more flexible transport, but that traditional Penning traps with a single trapping site (that confines a disc of ions) are not amenable to quantum computing. Hence the motivation for this work on microfabricated Penning traps, which have greater positional control of ions.

B. Based on a literature search (though perhaps not exhaustive) it seems like this is the first demonstration of full quantum control in a microfabricated Penning trap, and also one of the earliest demonstrations of a microfabricated Penning trap (following up on the author's previous demonstration of trapping in 2022). If so this is original and significant, since microfabrication is the most promising path to scalable quantum computing, and because the Penning trap has some nice advantages relative to the more common Paul traps that are used for QC.

C. The data and methodology presented in the paper comprehensively supports the authors' claims. It is clear and high quality, especially the motional heating rates. Given the enduring mysteries around "anomalous heating" in ion traps, these measurements provide a useful data point in understanding that phenomenon.

D. Yes; the data described in the paper as well as that shown in figures 2 and S1 have appropriate errors and fits.

E. The main conclusion of the paper is that full quantum control can be performed with an ion in a microfabricated Penning trap. This is well supported. I do think the paper would benefit from elaborating a bit more on the anticipated (if any) challenges of applying two-qubit gates in the envisioned scheme, especially with respect to the magnetron and cyclotron modes (assuming the gate is applied on the axial mode. These radial modes are geometrically different from those in a

Paul trap, though perhaps if they are cold they have the same negligible impact on an axial gate as one performed in a Paul trap. Those readers that are more familiar with the Paul trap modes will benefit from a few sentences of comparison.

F. A few items:

-The point about RF power dissipation is certainly valid at a high level, and the advantage of Penning traps vs Paul traps is clear in this regard, but it is an overstatement to say that it is “a very severe constraint in trap arrays of more than 100 sites”. In reference 21 for instance the estimate is that each ion site would contribute 2 μ W of RF power dissipation, which would be negligible. That said, this is likely an underestimate; [Blain et al in QST 6 034011] estimate higher power dissipation that accounts for RF lead lengths and dielectric loss. Even with higher dissipation values it would be more fair to call this a challenge that needs to be solved in order to scale, but it hasn’t really been necessary to address this to date and there are clear and reasonable approaches (like fanout of RF on low-loss packages, or employing techniques for lowering capacitance). Also reference 22 mostly addresses power dissipation due to DACs, not RF (apologies if I missed the RF part). In a nutshell I think the RF power dissipation motivation for Penning traps should be noted as a challenge for Paul traps and that Penning traps have a clear advantage, but the authors should not describe it as a “very severe constraint” that limits a trapped ion QC to a particular ion number. Or if they want to keep that assertion in they should more specifically support it.

-The ion lifetimes are quite good, what is the potential depth of the trap?

-Is the size of a 2d array of ions limited by the “10s of micrometres” difference in DC and RF null locations that can be tolerated during the axialization enabled Doppler cooling?

-I can think of two challenges with leveraging the greater ion density of a Penning trap compared to an RF Paul trap:

1)The control electrodes that would move the ions around need to create multiple harmonic potentials with nulls that determine the ion positions. Given that the ion-to-surface distance sets the natural size scale of the electrodes, how can a dense 2d array of ions (with individual control of the ions) be generated? This isn’t that much different than the same constraints for splitting and joining ions in the linear direction in a Paul trap, except in 2 dimensions.

2)If integrated optics are going to be delivering light to ions in a dense array, the output gratings will have dimensions also on the same scale as the ion height. This limits the density of optically controlled sites to a much lower value than the density of ions. If that is the case, how can one take advantage of a dense array of ions?

-I don’t think either of these questions needs an answer with specific design details, just a general concept or idea would be useful.

-I suggest adding the RF filter values to figure S4.

G. Yes, the references are appropriate (see F for a suggestion).

H. The manuscript is well written and clear.

Author Rebuttals to Initial Comments:

Referee #1 (Remarks to the Author):

I think this is an excellent article that reports a significant advance in the field of quantum information processing with trapped ions. This is not the first time that a single ion in a Penning trap has been trapped, sideband cooled and manipulated using quantum control techniques, but it is the first time that this has been done with a microfabricated Penning trap that has the potential for scaling to large numbers of ions by fabricating an array of such traps. The trap design is elegant and novel, and the techniques that are used are advanced and highly effective. This work represents a very significant advance on earlier work with ions in a Penning trap. The low motional excitation after sideband cooling in 3-d and the very low heating rates demonstrated are impressive and advance significantly on previous work.

The article is clear and informative. It has an appropriate set of references. The data is convincing and the figures are helpful. The methods are sound.

We thank the referee for their thorough reading of the manuscript and for identifying its novel features.

I have one significant criticism. The authors claim that they demonstrate "the ability to transport the ion arbitrarily in the trapping plane" (abstract). They claim that figure 3 demonstrates shuttling of the ion. My understanding of "shuttling" would be moving an ion between different trapping zones and this is a requirement for quantum information processing using trapped ions, as described by the authors on page 1. What the authors have done here is to move an ion around within one trapping zone, with a maximum excursion of about 40um (figure 3). This type of movement is not much more advanced than adjusting the position of the ion in order to minimise micromotion, which is a commonly-used technique (e.g. line 113). The range of movement demonstrated here is not much more than the width of one electrode. I think that this claim is therefore too strong and is not justified by the results presented. In order to be able to claim arbitrary transport of an ion, the authors should be able to show that they have moved the ion between different trapping zones.

In the abstract it is not clear what they mean by "This unique feature..". If this relates to their claimed ability to transport the ion, it is unjustified. What is it that's "unique"? Transport in 2-d has been demonstrated many times with microfabricated RF traps. Where have they demonstrated "improved connectivity and flexibility" in this paper?

In line 23 the authors imply that their Penning trap design either eliminates the need for specialized junctions or makes them easier to construct, and I do not believe that this is the case. In this paper they don't even discuss how to move an ion around on a quantum CCD, let alone through junctions. If they want to make this claim, I think the justification needs to be clearly explained.

Since the magnetic field is uniform across the whole chip, the trap site is formed wherever the null of the dc field is placed. The dc null need not be aligned to the null of another potential (as is the case in an rf trap) while this transport is being done. By moving this dc null from any point A to any point B in a straight line, we can transport the ion directly. In fact, this is how Fig 3 is generated – the shuttling is not decomposed into two perpendicular movements along the x and z axes, but rather it is along the direct path joining points A and B. In this sense, the transport is arbitrary. Since the ion does not need to be moved around a corner as in rf traps, we the need for junctions is eliminated

and the trap design is significantly simplified. We have now emphasized that no junctions are required in the text.

We have transported ions by 250 microns in the current trap, however it did not occur to us that this was more worth reporting than what was shown already, since the ease of transporting ions in the trap meant that we didn't anticipate additional problems. The main limitation in what was shown was the actual imaging in the laser beam, not the transport. To show to the referee that we can transport further than 40 microns we include images below. These show that we can transport further beyond what one might consider as the limits of a 'zone'. However, we don't think the main text benefits from the inclusion of this data.

Fig 1: An ion is transported from the original location at 152 μm trap height (bottom left image) to a location roughly 100 μm away along the axial direction z (top left image), as well as along the radial in-plane direction x (bottom right image). The imaging system used for taking these pictures had a total magnification of approximately 31, while a pixel is 16 μm wide. The measured distances of 195 (191) pixels thus correspond to 100.6 (98.5) μm .

In line 54, the static array references should include Crick et al, Rev Sci Instrum 81, 013111 (2010), where shuttling between different trapping sites was demonstrated in a Penning trap. This discussion about ions in separate sites being brought together, and arbitrary transport in 3-d, is way beyond what is demonstrated in this paper.

The statement the referee makes is not accurate. Crick et al. demonstrated transport of an ensemble of ions between two discrete positions, using a non-adiabatic transport technique. On each transport forward and back, they lost a good fraction of these ions, which would be intolerable for quantum computing. It is worth noting that such non-adiabatic techniques also require exquisite timing of voltage control when operated at suitably high trap frequencies for performing high fidelity quantum

operations. In our demonstration we transport the ion to any location in a 2-D plane, never lose it, and furthermore do this with the ion in the ground state. Thus we think our work exceeds in many ways the results of Crick et al. We added a reference to Crick et. Al. in line 285, where we have added a comparison text: "This contrasts with earlier non-adiabatic radial transport of ensembles of ions in Penning traps, where a good fraction of the ions were lost in each transport"

As discussed above, the claim in lines 69-72 that they have demonstrated the "key ingredient for scaling" is not justified in my view. This may be possible in the future but there are significant challenges to overcome first.

See above for reply on transport. We think that this is certainly a key ingredient for scaling, albeit not the only ingredient for scaling. We realize that the wording of the previous statement "the key ingredient" was incorrect and have now changed this to "a"

In figure 1a it would be useful to give an indication of scale.

We have done so now.

In line 138 it would be useful to know how many photons are detected before the ion decays to the dark level. How long does this take?

The detection window is 300 microseconds, and it is estimated that the probability of off-resonantly exciting to an excited state and subsequently decaying to the dark state during this detection time is less than 10^{-5} .

In line 165 it would be helpful to define what is meant by a "single experiment" and to state what that timescale is. Also, what is the repetition rate of experiments?

We have clarified the statement to read: "The nature of the noise appears to be slow on the timescale (~10 ms) of a single experimental shot consisting of cooling, probing and detection and the fringe contrast decay follows a Gaussian curve." The repetition rate can thus be around 100 Hz.

In the caption of figure 2 I suggest including the axial frequency for the experiment with the trap detached in the caption, to save the reader having to look elsewhere in the paper.

The manuscript has been edited to include this.

The statement in line 276-9 is not very convincing as the ion is only Doppler cooled here. A more convincing and sensitive demonstration would be moving a ground-state cooled ion from one place to another and then demonstrating no heating of the motion during this process.

We agree with the referee. Although we don't have space to add data in the main text, we have added a more concrete statement regarding near ground state transport of an ion, and included the data to back it up in the supplementary information. We observe no additional heating beyond the ambient heating for an un-transported ion when we introduce the transport.

A small point about the references: there are many examples of missing capital letters (especially for Penning) and lost formatting of super- and sub-scripts.

We have included the correctly typeset bibliography file with our manuscript to the editors and as far as we can tell, the format is enforced by the REVTeX formatting that was chosen for our arXiv submission of the manuscript. We defer to the nature copywriting team to set the appropriate formatting of the bibliography.

I find the exceedingly brief statement in lines 620-621 and the two-line caption inadequate for Fig S1. The authors should explain what the data is showing and how it was achieved. Is part b a plot of fringe visibility or population?

The caption has been edited to include more details on the data shown and the Uhrig dynamical decoupling sequence.

In line 631 it is frustrating that they state there are 25 electrodes but we can only see 21 in figure 1. Why not show them all?

The choice was made in the interest of space, and to make the middle electrodes (which are less wide) more prominent.

Somewhere it would be good to state how the photo-ionization is achieved. Which lasers are used and what is the atomic process?

We have edited the manuscript to include the wavelength of the laser (235 nm) used for resonance-enhanced multiphoton ionization.

Referee #2 (Remarks to the Author):

* Summary of the manuscript

Typical approach for the construction of quantum computer (QC) with trapped ions utilizes micro-fabricated Paul trap which requires rf fields with high voltage. So far, quantum computation and quantum simulation with tens of ions (qubits) based on this approach have been experimentally demonstrated already, and they even lead to many commercialized systems. However, to increase the number of qubits beyond the current scale, several limitations should be addressed.

Some of the remaining challenges are the significant power dissipation originating from the large rf voltages necessary for all Paul traps, and the accompanying micromotion which requires frequent re-alignment of the DC null against the rf null. The authors claims that the requirement of rf fields can be resolved by using a Penning trap which mainly requires static magnetic and electric fields only, and they demonstrated that a single 9Be^+ ion can be trapped in a micro-fabricated Penning trap, and three motional modes of this ion can be cooled down very close to the ideal ground states. One of the challenges in reaching the motional ground states of the Penning trap is the Doppler cooling of the magnetron mode with a negative quantized energy, and such mode has been efficiently cooled by the axialization process that mixes the magnetron mode and the cyclotron mode with small amplitude of rf voltage, and then through an additional sideband cooling. They also characterized the coherence time of axial modes and the internal electronic state which will be used as qubit. The measurement result also showed that the significant part of the motional dephasing comes from the electrical noise originating from the external electronics, especially for the magnetron mode having relatively lower frequency. Finally, they also claim that the QCCD architecture originally devised for the micro-fabricated Paul trap can be extended to the micro-fabricated Penning trap by demonstrating the adiabatic shuttling of a single ion, which in principle has more degrees of freedom compared to the Paul trap where shuttling is considered only along the rf rails.

The main novelty of this work is that the authors demonstrated many of components necessary to build scalable quantum computing systems with trapped ions using Penning trap, and provided convincing evidences that micro-fabricated Penning trap might be a strong alternative to the current Paul trap approach. They even devised a novel method to detach the external voltage sources in the middle of quantum operations, which can allow low-noise operation and measurement and might even find applications in other systems. On the other hand, this work did not demonstrate any components required for multi-qubit operation such as trapping of the controlled numbers of ions, combining and splitting of them in the surface trap, and quantum gate between two qubits etc. I should admit that implementation of these components will require fair amount of additional work and each of them can even lead to a separate publication, and therefore it might be still too much to require results of any of these operations in the current manuscript. However, this also might mean that implementation of these components will be non-trivial for the experiments. In such a case, I think it will be fair for the authors to elaborate more about the remaining challenges to demonstrate the multi-qubit operations in the conclusions.

I find that the overall manuscript is well-written and what has been done experimentally is properly presented. The authors used standard method to estimate the number of phonons and heating rate in each motional mode, and Ramsey interference has been utilized to measure the coherence time of both spin and motional mode while dynamic decoupling methods have been applied to increase the

coherence time. Therefore, all the included data and methodology are valid for the purpose of the experiments. Also, all the relevant references are properly included.

We would like to express our gratitude to the referee for carefully reviewing our manuscript and for recognizing its unique elements. We are of the same opinion as the referee, in that, characterizing the setup beyond a single qubit – for instance, measuring the fidelity of a two-qubit gate – seems beyond the scope of this paper; in particular due to the format and word limit of a Nature article. While we would not consider the implementation of multi-qubit entanglement as trivial, at the same time we expect that the standard techniques used for an MS gate would work equally well in our micro-fabricated Penning trap as it would for a typical rf trap or large-scale Penning trap. For a future publication we will certainly investigate applying an MS gate, for example on the COM mode of a linear chain where we expect the spin-motion dynamics to be identical to the case of an RF trap. We have now added a sentence in the outlook stating that multi-qubit gates would follow standard methods demonstrated in rf traps, and included citations to the multi-well gate performed at NIST, and to our theory paper, which describes how to implement multi-qubit gates.

* Major concerns

- None

* Minor concerns

- I wonder whether the expression of "unit cell" in the title is appropriate. "Unit cell" can imply that the entire system can be constructed only by replication of the unit cells, but given that the multi-qubit operations are missing, the title might be misleading.

The intention behind the use of the term 'unit cell' is indeed to reflect the fact that a 2-d grid of ions can be achieved by tiling a subset of electrodes. Each subset provides the voltages for creating a single trapping site, each containing a single ion. Nevertheless, we have adjusted the title to "Penning micro-trap for quantum computing" to more straightforwardly state the results of the paper.

- Similar to the above point, the claim that "we demonstrate full quantum control of an ion" in the abstract needs some contemplation, because "full quantum control of an ion" might mean that the quantum state of the motional modes is also fully controlled such as the implementation of the state dependent force or generation of Schrodinger cat state as shown in PRL 116, 140402 (2016) by the same group, but this work mostly includes partially incoherent process such as sideband cooling and phonon number estimation. Of course, the sideband Ramsey experiment in Fig. 2 (c) might be considered as coherent control of the motional mode to some degree, but with the coherent control, there should be more characterization of the fidelity of the operation. Also, people will be more interested in the application of these states to multi-qubit gates, and given that the two-qubit operations are missing, the use of "full quantum control of an ion" can be misleading.

The coherence measurements on the spin and motion of a single ion require coherent control of a single ion. The operations used form a universal set for both spin and motion. As a result, we consider that we have demonstrated full quantum control over the degrees of freedom of a single ion. We agree that we have not demonstrated multi-qubit operations, but since the trap operates with excellent performance, and this is the main change compared to a Paul trap, we see no reason

why all of the operations which are standard in Paul traps should be significantly worse in the Penning trap. As noted above, we think that demonstrating multi-qubit gates would be an additional step which goes beyond the scope of the current paper.

- Considering the ion height of the 152 μ m and the axial frequency of 2.5 MHz, I am a little worried about the necessary DC voltages for the axial confinement, but I could not find the voltage information in the manuscript. Especially, in the conclusion, the authors claim that the fabrication of the planar Penning trap will be compatible with the CMOS fabrication process, but if the necessary DC voltages are in the range of tens of volts, it might not be compatible with CMOS foundry. Therefore, I want the authors to include more information about the DC voltages. Also, as the authors mentioned in the conclusion, incorporation of the integrated optics for scalability might require use of heavier ions such as Sr⁺ or Ba⁺, and I am afraid that other challenges might creep up as higher DC voltage might be necessary to maintain the similar axial frequency. Especially, for the two-qubit operation, if they plan to apply schemes based on the axial phonon modes, I believe that they want to keep the axial frequency high as in the current setup, but I might be wrong. I hope the authors give more comments on the use of heavier ions.

Currently, to achieve an axial frequency of 2.5 MHz at 152 μ m, we require a maximum voltage of approximately 20 V. Since an equivalent frequency with heavier ions like strontium and barium would require a very high magnetic field (at least 10 T) to begin with, we envision going to calcium to utilize the integrated optics. The reference 'Ion Traps Fabricated in a CMOS Foundry' by Mehta et al. suggests that a dc voltage of 200 V can be applied to typical CMOS-based traps, allowing us to work with calcium at with ease. Nevertheless, the reviewer is correct in pointing out that we might need to work at a lower ion-electrode distance in order to work with heavier atoms.

- In the conclusion, coherence time of spin can be increased by the improved mechanical stability of magnet, but there is no comment on the fluctuation of the current. Could you please include estimated error caused by the mechanical instability and the fluctuation of the current?

If we assume that the magnetic field fluctuations purely result from fluctuations in the current through the coils, we estimate a standard deviation of 42 nA. It is however not very clear at the moment what the exact sources of the field instability are, and what their relative contributions are.

- To generate Fig. 3, the ion was Doppler cooled at the original location, but there is no information about this location. Please indicate where this original location is in Fig. 3. Also, Fig. 3 shows thicker horizontal lines compared to the slightly tilted vertical lines, and I wonder if this is caused by visual artifact, or there is any physical explanation.

- Electrical detachment of the trap from the external voltage source seems to be a useful trick for other experiments, and many readers might find useful if you can include the model number of the MOSFET switch.

The manuscript has been edited to include the part number.

* Conclusions

The experimental results of this study provide significant evidence that the micro-fabricated Penning trap might open a new avenue towards scalable quantum computing architecture with trapped ions, even though it seems that there remain many challenges. And, publication of this work can motivate

more researchers to attack these challenges, and expedite the overall progress in the development of quantum computer based on ion trap.

Referee #3 (Remarks to the Author):

A. This manuscript describes the operation of a microfabricated Penning trap and full quantum control of the ion in the trap. The authors correctly note that Penning traps have advantages over Paul traps in terms of reduced voltage requirements, lower power dissipation, ion density, and more flexible transport, but that traditional Penning traps with a single trapping site (that confines a disc of ions) are not amenable to quantum computing. Hence the motivation for this work on microfabricated Penning traps, which have greater positional control of ions.

B. Based on a literature search (though perhaps not exhaustive) it seems like this is the first demonstration of full quantum control in a microfabricated Penning trap, and also one of the earliest demonstrations of a microfabricated Penning trap (following up on the author's previous demonstration of trapping in 2022). If so this is original and significant, since microfabrication is the most promising path to scalable quantum computing, and because the Penning trap has some nice advantages relative to the more common Paul traps that are used for QC.

C. The data and methodology presented in the paper comprehensively supports the authors' claims. It is clear and high quality, especially the motional heating rates. Given the enduring mysteries around "anomalous heating" in ion traps, these measurements provide a useful data point in understanding that phenomenon.

D. Yes; the data described in the paper as well as that shown in figures 2 and S1 have appropriate errors and fits.

E. The main conclusion of the paper is that full quantum control can be performed with an ion in a microfabricated Penning trap. This is well supported. I do think the paper would benefit from elaborating a bit more on the anticipated (if any) challenges of applying two-qubit gates in the envisioned scheme, especially with respect to the magnetron and cyclotron modes (assuming the gate is applied on the axial mode. These radial modes are geometrically different from those in a Paul trap, though perhaps if they are cold they have the same negligible impact on an axial gate as one performed in a Paul trap. Those readers that are more familiar with the Paul trap modes will benefit from a few sentences of comparison.

We want to thank the referee for taking the time to review our manuscript thoughtfully and for acknowledging its special qualities. To our knowledge as well, we are the first group to work with Penning traps of such a scale. We expect that the techniques for demonstrating multi-qubit operations in rf traps or larger-scale Penning traps will apply to our system as well, especially if the interactions are mediated via the axial modes. For interaction via the radial modes, one distinctive property is that addressing only one of the modes by selecting the direction of the laser propagation is not possible. As a result, the fidelity may benefit from cooling of both radial modes prior to the entanglement operation. However, this distinction is a minor technicality; the 'circular' nature of the mode eigenvectors does not factor into how the gates themselves would be performed.

F. A few items:

-The point about RF power dissipation is certainly valid at a high level, and the advantage of Penning traps vs Paul traps is clear in this regard, but it is an overstatement to say that it is "a very severe constraint in trap arrays of more than 100 sites". In reference 21 for instance the estimate is that

each ion site would contribute 2 μW of RF power dissipation, which would be negligible. That said, this is likely an underestimate; [Blain et al in QST 6 034011] estimate higher power dissipation that accounts for RF lead lengths and dielectric loss. Even with higher dissipation values it would be more fair to call this a challenge that needs to be solved in order to scale, but it hasn't really been necessary to address this to date and there are clear and reasonable approaches (like fanout of RF on low-loss packages, or employing techniques for lowering capacitance). Also reference 22 mostly addresses power dissipation due to DACs, not RF (apologies if I missed the RF part). In a nutshell I think the RF power dissipation motivation for Penning traps should be noted as a challenge for Paul traps and that Penning traps have a clear advantage, but the authors should not describe it as a "very severe constraint" that limits a trapped ion QC to a particular ion number. Or if they want to keep that assertion in they should more specifically support it.

We agree with the referee that perhaps the emphasis was too strong. We have removed the final sentence of the paragraph, which emphasized this point, and also added the Blain et al reference in the previous sentence.

-The ion lifetimes are quite good, what is the potential depth of the trap?

The potential depth in the axial direction is 0.82 eV at 2.5 MHz trap frequency. In the radial plane, the peak of the potential is at 4.9 eV though the potential becomes severely anharmonic much earlier leading to inability to cool or particle loss. A more empirical definition of the trap depth can be obtained by solving the equations of motions in the simulated potential for a particle injected along the B axis and seeing up to what initial energy stable trajectories are supported. We find this to be around 0.1 eV.

-Is the size of a 2d array of ions limited by the "10s of micrometres" difference in DC and RF null locations that can be tolerated during the axialization enabled Doppler cooling?

We think that this should not be a limitation. In a 2-D array it should be possible to design electrode structures (as for rf trap arrays) in which there are a number of nulls of the rf field. Our current trap does not allow this, in part due to co-wiring of electrodes which we performed when trying to load the trap for the first time.

-I can think of two challenges with leveraging the greater ion density of a Penning trap compared to an RF Paul trap:

1)The control electrodes that would move the ions around need to create multiple harmonic potentials with nulls that determine the ion positions. Given that the ion-to-surface distance sets the natural size scale of the electrodes, how can a dense 2d array of ions (with individual control of the ions) be generated? This isn't that much different than the same constraints for splitting and joining ions in the linear direction in a Paul trap, except in 2 dimensions.

2)If integrated optics are going to be delivering light to ions in a dense array, the output gratings will have dimensions also on the same scale as the ion height. This limits the density of optically controlled sites to a much lower value than the density of ions. If that is the case, how can one take advantage of a dense array of ions?

As detailed in our previous publication (see Jain et al. *Scalable arrays of micro-Penning traps for quantum computing and simulation*, Phys. Rev. X 10, 031027 (2020)), packing the nulls of the combined pseudopotential in an array of rf traps requires a greater voltage as compared to an array of Penning traps. In other words, for a given voltage, the distance-to-height ratio can be much higher. To answer point 2, the size of the grating is contingent on how tightly focussed the beams need to

be. If the two ions are in two separate wells 30 micrometres apart, the grating size would be in the range of 5 μm to 10 μm (see Beck et al. *Grating design methodology for tailored free-space beam-forming*, arXiv:2306.09220). While the footprint of the gratings should rightfully be considered when thinking of the grid spacing we could achieve, we do not think it would be the limiting factor.

-I don't think either of these questions needs an answer with specific design details, just a general concept or idea would be useful.

-I suggest adding the RF filter values to figure S4.

The manuscript has been edited to include the resistance and capacitance of the RC filter.

G. Yes, the references are appropriate (see F for a suggestion).

H. The manuscript is well written and clear.

Reviewer Reports on the First Revision:

Referees' comments:

Referee #1 (Remarks to the Author):

The authors of this paper have given a helpful and detailed response to all three referees' comments. They have provided additional explanations and some new data. I think the paper has been improved significantly by the changes that the authors have made.

I do not have any remaining issues.

Referee #2 (Remarks to the Author):

I think the authors clarified most of the questions raised by all the reviewers and made the proper changes in the manuscript. Especially the addition of the data in Fig. S2 seems to support their original argument that transport of the ion did not increase the heating rate.

Therefore, I recommend the publication of the manuscript with a minor request:

I asked the following questions, and the authors seem to have missed it: In Fig. 3, could you please indicate the first location where the ion is Doppler-cooled before it is transported to the 2nd desired location? Also can you explain why the horizontal lines in Fig. 3 look thicker than the slightly tilted vertical lines? Is it possible that different position resolutions are used for the horizontal coordinate compared to the vertical coordinate when the 58 positions are determined? Or it can be just a visual artifact, but it might be related to different heating rates along different axes. However, with the limited information, I cannot draw any conclusion.

Referee #3 (Remarks to the Author):

I am satisfied by the authors' response to my questions and comments, and recommend this paper for publication.

Author Rebuttals to First Revision:

Referee #2 (Remarks to the Author):

I think the authors clarified most of the questions raised by all the reviewers and made the proper changes in the manuscript. Especially the addition of the data in Fig. S2 seems to support their original argument that transport of the ion did not increase the heating rate.

Therefore, I recommend the publication of the manuscript with a minor request:

I asked the following questions, and the authors seem to have missed it: In Fig. 3, could you please indicate the first location where the ion is Doppler-cooled before it is transported to the 2nd desired location? Also can you explain why the horizontal lines in Fig. 3 look thicker than the slightly tilted vertical lines? Is it possible that different position resolutions are used for the horizontal coordinate compared to the vertical coordinate when the 58 positions are determined? Or it can be just a visual artifact, but it might be related to different heating rates along different axes. However, with the limited information, I cannot draw any conclusion.

The location where the ion is Doppler cooled is now indicated by a red circle in the figure. We suspect that the horizontal lines look thicker because of how the image of a single ion looks, which itself is probably a result of aberrations caused by misalignment of the objective with respect to the imaging axis. Here is a sample image of a single ion: